

# Optimizing e-commerce warehousing through open dimension management in a three-dimensional bin packing system

Jianglong Yang[1,*], Kaibo Liang[2,*], Huwei Liu[1], Man Shan[1], Li Zhou[1], Lingjie Kong[1] and Xiaolan Li[1]

[1] School of Information, Beijing Wuzi University, Beijing, China
[2] School of Management and Engineering, Capital University of Economics and Business, Beijing, China
[*] These authors contributed equally to this work.

Corresponding author
Man Shan,
22311087100015@bwu.edu.cn

## ABSTRACT

In the field of e-commerce warehousing, maximizing the utilization of packing bins is a fundamental goal for all major logistics enterprises. However, determining the appropriate size of packing bins poses a practical challenge for many logistics companies. Given the limited research on the open-size 3D bin packing problem as well as the high complexity and lengthy computation time of existing models, this study focuses on optimizing multiple-bin sizes within the e-commerce context. Building upon existing research, we propose a hybrid integer programming model, denoted as the three dimensional multiple option dimensional rectangular packing problem (3D-MODRPP), to address the multiple-bin size 3D bin packing problem. Additionally, we leverage well-established hardware and software technologies to propose a 3D bin packing system capable of accommodating multiple bin types with open dimensions. To reduce the complexity of the model and the number of constraints, we introduce a novel assumption method for 0–1 integer variables in the overlap and rotation constraints. By applying this approach, we significantly streamline the computational complexity associated with the model calculations. Furthermore, we refine the dataset by developing a customized version based on the classical Three-Dimensional One-Size Dependent Rectangular Packing Problem (3D-ODRPP) dataset, leading to improved outcomes. Through comprehensive analysis of the research results, our model exhibits remarkable advancements in addressing the strong heterogeneous bin packing problem, the weak heterogeneous bin packing problem, the actual bin packing problem, and the bin packing problem with multiple bin types and open sizes. Specifically, it significantly reduces model complexity and computation time and increases space utilization. The system designed in this study paves the way for practical applications based on the proposed model, providing researchers with broader research prospects and directions to expand the scope of investigation in the field of 3D bin packing. Consequently, this system contributes to solving complex 3D packing problems, reducing space waste, and enhancing transportation efficiency.

## INTRODUCTION

With the exponential growth of e-commerce, the surging consumer demand has resulted in an overwhelming number of express parcels. While providing convenience to people's daily lives, the accumulation of express packing waste poses a significant threat to the natural environment. According to data from the China Post Office, the express delivery industry in China has been handling an average of 180 million deliveries per day (*China Post Office, 2022.*). In 2022 alone, the total volume of express business reached a staggering 110.58 billion pieces, marking a 2.1% increase compared to the previous year. Furthermore, by June 30, 2022, the national express business volume had already surpassed 51.22 billion pieces, exhibiting a notable year-on-year growth rate of 3.7%. Astonishingly, as of April 2023, China's express business volume had already reached a remarkable milestone of 37.1 billion pieces, demonstrating a substantial year-on-year growth rate of 17.0% (*China Post Office, 2023*). On the one hand, this highlights the remarkable progress of e-commerce logistics, underscoring its rapid development. However, on the other hand, it is imperative to address the pressing concern of wasteful and environmentally unfriendly packing practices (*Zhao et al., 2022b*). An increasing number of e-commerce companies are embracing the use of recyclable packing bines as part of their commitment to minimizing environmental impact. Furthermore, certain logistics packing companies are taking measures to enhance downstream logistics and distribution efficiency by standardizing the form and size of packing bines, ensuring uniform packing specifications that are easy to handle. By employing regular items, logistics space resources can be maximized, and there is even a growing trend for the entire industry to adopt similar or identical packing standards in order to reduce costs. Hence, finding the optimal bining solution that emphasizes the efficient use of packing materials and environmental consciousness becomes crucial. Only by pursuing such an approach can we achieve sustainable development while simultaneously enhancing logistics efficiency.

The bin packing problem can be divided into several dimensions: one-dimensional, two-dimensional, and three-dimensional. The three-dimensional packing problem encompasses both the one-dimensional and two-dimensional packing problems (*Elhedhli, Gzara & Yildiz, 2019*). Additionally, packing problems can be categorized into two types based on the number of packing bins: single-bin packing problems and multi-bin packing problems. Furthermore, the packing problem can be further subdivided based on the variability of the packing bin size: single-bin size fixed problem, single-bin size variable problem, multiple-bin size fixed problem, and multiple-bin size variable problem. Currently, most of the research in this field primarily focuses on the single-bin packing problem and the multiple-bin size problem. However, there is relatively less research on the multiple-bin size variable problem, which is characterized by high model complexity and long calculation times.

In this article, we aim to tackle the challenge of optimizing the utilization of multiple-bin sizes commonly encountered in the logistics industry. Specifically, we focus on efficiently arranging various bin sizes within a given space to maximize capacity. To address this problem, we designed a three-dimensional bin packing system for multiple bin sizes. The

system consists mainly of a hybrid integer programming model for the 3D-MODRPP and the simulation of the actual bin packing problem using existing techniques. Our system primarily comprises a hybrid integer programming model for the 3D bin packing problem with 3D-MODRPP and a simulation of the actual bin packing problem using established techniques. By evaluating the outcomes of our model, we have observed significant reductions in both model complexity and computation time compared to existing methods. Moreover, our system aligns well with real-world bin-packing scenarios, thus proving its practical applicability. The implications of our work extend beyond the logistics field. Not only can our system be widely adopted in logistics operations, but it also serves as a valuable reference for addressing similar problems in various domains. By presenting innovative approaches and tangible results, our research contributes to the advancement of optimization techniques in this field.

This article addresses a significant research gap within the realm of e-commerce warehousing, specifically in the context of solving the multifaceted challenge of the bin-type three-dimensional packing problem. Presently, research in this domain is considerably limited, highlighting the need for innovative solutions to counter the intricate nature of existing models and the prolonged computational time associated with them. By introducing an inventive 0–1 integer variable assumption method and an optimization model, we have successfully tackled the intricate 3D packing problem encompassing various bin types and their open dimensions. This achievement serves as an effective remedy for real-world logistics operations. The model and methodology proposed in this article effectively streamline the resolution of the multi-dimensional packing problem by multiple bin sizes, exhibiting a marked reduction in both model intricacy and computation time when juxtaposed with existing methodologies. This reduction will inevitably enhance the efficiency and curtail operational expenses for logistics enterprises during their real-world endeavors. Furthermore, this study contributes to the optimization of bin dimension design, thus enabling the maximization of bin space utilization and the curbing of resource wastage. Consequently, warehouse space occupation will be minimized while resource utilization is optimized. Through the optimized 3D packing scheme, the article significantly aids in the volume reduction of goods, leading to diminished space wastage during transportation and improved loading efficiency. This, in turn, culminates in enhanced overall transportation efficiency. The 3D packing system is tailored to accurately replicate real-world scenarios, making it applicable not only across logistics domains but also as an inspiration for researchers to venture into more intricate packing predicaments, thus propelling the advancement of this field.

The remaining sections of this article are structured as follows: 'Literature Review' provides a comprehensive analysis of the existing studies conducted by researchers on the 3D bin packing problem. This section delves into their findings and insights, offering a thorough examination of the current state of research in this field. Moving forward, 'Problem Description' presents a detailed description of the multiple-bin-type 3D bin packing problem. This section offers a comprehensive overview of the problem, outlining its intricacies and complexities. In 'Variable Assumptions and Model Construction', we introduce a novel hybrid integer programming model specifically designed for addressing

the multiple-bin-type 3D bin packing problem, which we refer to as 3D-MODRPP. This model is meticulously described and developed, showcasing its potential to tackle this challenging problem effectively. To validate and assess the performance of our proposed algorithm, 'Computational Experiments and System Design' provides an in-depth evaluation based on existing datasets. This section highlights the algorithm's efficacy and provides an analysis of its results, further reinforcing its practicality and potential applications. 'Discussion' presents the theoretical and practical implications of the findings of this article. Finally, 'Conclusions and Prospect' offers concluding remarks summarizing the key findings and insights derived from this study. Additionally, this section explores potential avenues for future research, emphasizing areas that warrant further investigation and development in the field of 3D bin packing.

## LITERATURE REVIEW

In the field of e-commerce stores, the three-dimensional packing conundrum pertains to efficiently arranging items of various shapes within limited three-dimensional containers. The goal is to minimize the space occupied by the items while adhering to specific constraints, such as preventing item overlap and adhering to restrictions on item placement directions. The packing problem can be categorized as either a single-bin type or a multiple-bin type, depending on the number and type of containers involved. In the bin of single-bin type packing, it can further be classified as either a fixed-size or variable-size problem based on whether the bin size remains constant or not. Similarly, the multiple-bin-type three-dimensional packing problem can be divided into a fixed-size problem or a variable-size problem, depending on whether the sizes of the different types of containers are fixed or subject to change.

### Single-bin 3D packing problem with fixed dimensions

For the single-bin 3D packing problem with fixed dimensions, researchers have primarily focused on studying it through heuristic algorithms to address the challenge of solving large-scale 3D packing problems within a reasonable timeframe. *Mahvash, Awasthi & Chauhan (2018)* employed a heuristic algorithm based on the column generation (CG) technique to tackle the single-bin 3D packing problem. *Di Puglia Pugliese, Guerriero & Calbi (2019)* developed a simple and effective heuristic approach to address a 3D bin packing problem that considers various operational constraints with the goal of inserting objects with the same destination into a new container. Recently, *Yang et al. (2023)* investigated the applicability of heuristics for online 3D binning problems and explored how to effectively integrate heuristics with deep reinforcement learning techniques. It should be noted that, given the nature of 3D packing studies, heuristic algorithms do not guarantee to find the globally optimal solution. Therefore, researchers often resort to small-scale exact algorithms such as column generation algorithms and branch-and-bound algorithms. *Elhedhli, Gzara & Yildiz (2019)* proposed a novel formulation and a column generation-based solution for solving the 3D binning problem. Similarly, *Silva & Wauters (2019)* focused on exact methods for comparing 3D cutting and packing problems. They adapted the chosen method to the specific problem analyzed and conducted experiments using classical benchmark

datasets as well as newly generated examples from cutting and packing generators in the literature, enabling a comprehensive comparison.

Furthermore, *Borges et al. (2020)* proposed two dynamic programming algorithms and a branch-and-bound algorithm to address the 3D binning problem. These algorithms aim to find optimal solutions by considering different constraints and variations within the problem. *Gzara, Elhedhli & Yildiz (2020)* employed a layer-based column generation method to enhance the loading rate of a single-bin 3-D packing problem. *Erbayrak, Özkır & Yıldırım (2021)* employed a weighted objective function to determine the optimal packing solution that minimizes the number of bins used while also balancing the deviation from the ideal center of gravity and maximizing the family uniform ratio. *Küçük & Yildiz (2022)* proposed a constraint planning-based solution to the three dimensional loading capacity vehicle path problem (3l-CVRP) in distribution logistics, which involves solving a combination of a vehicle route and a three-dimensional loading problem. *Tresca et al. (2022)* combined a mixed integer linear programming (MILP) formulation and layer construction heuristics to solve the single bin fixed-size 3D packing problem. *Tole et al. (2023)* proposed an algorithm consisting of two phases: an initialization phase and a refinement phase for the fixed-size single-bin 3D packing problem. A first-time fit grid search (FGS) algorithm to generate the initial solution, and then a simulated annealing (SA) algorithm was designed to explore the solution space of the initial solution by the proposed novel local neighborhood search strategy known as circular quadrant perturbation.

Several researchers have also combined algorithms to address the 3D container loading problem. *Huang et al. (2022)* introduced a novel technique called Ternary Search Tree Differential Evolutionary algorithm (TSTDE) to solve the 3D container loading problem (3D-CLP). *Zhao et al. (2022a)* proposed a novel stacking tree approach to optimize the packing strategy by analyzing the stability of the packing online. *Su et al. (2021)* combined a chemical reaction optimization algorithm with a greedy algorithm to solve the 3D bin packing problem, demonstrating superior performance compared to existing algorithms. *Shuai et al. (2023)* investigated the robotic three-dimensional bin packing problem (R-3DBP) and presented a new bin packing algorithm that considers sensor and planner machine errors, allowing self-correction of the position of boxed items. *Moura et al. (2023)* proposed a deep reinforcement learning (DRL) model utilizing the Transformer architecture as a policy network and approximate policy optimization (PPO) to train the network for solving the 3D packing problem. Finally, *Romero et al. (2023)* introduced a hybrid quantum–classical framework to tackle real-world 3D modeling problems.

## Open-size single-bin 3D packing issues

For the open-size single-bin 3D bin packing problem, several studies have been conducted to address different aspects of the problem. *Litvinchev, Pankratov & Romanova (2019)* investigated the challenge of packing a set of irregular objects into a rectangular container with variable height. They devised a solution algorithm that combines a fast starting point algorithm with an efficient local optimization process. *Araújo et al. (2019)* focused on additive manufacturing in the context of the irregular 3D bin packing problem. They introduced a new taxonomy and dataset using the 3D Irregular Items Problem (3DIP)

model algorithm. *Wang (2022)* undertook research on the online constrained variable size solving 3D bin packing problem, employing computer simulation techniques. *Harrath (2022)* explored the three-dimensional single bin size bin packing problem (3D-SBSPP) and proposed a three-stage heuristic algorithm. Their approach involved creating a minimum number of bins, allowing for six possible item rotation directions while ensuring that the crated items satisfy equilibrium constraints. The algorithm demonstrated superiority over other heuristics in various problem examples. *Mungwattana, Piyachayawat & Janssens (2022)* proposed a two-stage optimization method for solving the Dealer Pallet Loading Problem (DPLP) with multiple sizes. Their approach utilized an evolutionary algorithm and a differential evolutionary algorithm. The results indicated a 0.42% higher total number of pallets compared to the exact algorithm for the benchmark problem set, along with a significant 78.54% reduction in average computation time. In a recent study, *Que, Yang & Zhang (2023)* proposed a deep reinforcement learning model to tackle the 3D packing problem. They leveraged deep learning techniques to address the challenge of variable container height, enabling them to obtain superior solutions in shorter timeframes.

## Fixed size multiple-bin type three-dimensional bin packing problems

For the problem of packing multiple bins of different sizes into fixed-size 3D containers, researchers have predominantly relied on heuristic algorithms for their investigations. *Paquay, Limbourg & Schyns (2018)* introduced a fast constructive heuristic to address the 3D multiple different size bin packing problem. *Sangchooli & Sajadifar (2021)* developed a mathematical model for the 3D multi-bin packing problem and employed a novel heuristic algorithm along with the GRASP algorithm to solve it. *Li, Chen & Huo (2022)* proposed a hybrid adaptive large neighborhood search (HALNS) algorithm that encompasses a collection of primitive damage-repair operators and incorporates a heuristic packing algorithm specifically designed to tackle large-scale heterogeneous container loading problems within a limited time frame. *Zhao et al. (2022b)* conducted a study on the multi-carton 3D packing problem involving heterogeneous squeezable items in an e-commerce retail department store. They designed a heuristic algorithm aimed at selecting the most suitable packing materials to pack all items in an order, using the minimum number of packing materials while maximizing space utilization. *Trivella & Pisinger (2016)* used a multistage local search heuristic algorithm with the objective of packing objects of different sizes into a finite number of similar bins/containers to minimize the number of bins used. *Vieira et al. (2021)* investigated the problem of packing a wide range of bin sizes, and proposed a two-stage crating algorithm, which firstly uses a 0–1 plan to assign to the least number of bines, and then loaded into three-dimensional containers by mixed integer nonlinear planning to minimize the total container volume, and validated the effectiveness and efficiency of the method on data. *Alonso et al. (2019)* loaded onto trucks to solve the multiple container loading problem for companies providing services for customer orders and solved it by an Integer Linear Programming (ILP) model.

All of the aforementioned articles suggest the utilization of heuristic algorithms to address the 3D bin packing problem. While heuristics are relatively straightforward to implement, none of these approaches can guarantee global optimality for 3D bin packing. As a result,

*Li et al. (2022)* proposed a recurrent conditional query learning (RCLL) method to solve the 3D bin packing problem. *Hernández-Hernández & Castillo-García (2021)* proposed a fuzzy logic classification (FLC) approach to successfully assign bines to destinations that are as close as possible, addressing the 3D packing problem. *Dell'Amico, Furini & Iori (2020)* introduced a holistic algorithm that utilizes a hybrid approach to minimize the amount of packing required. *Agarwal et al. (2020)* presented a reliable and efficient bin packing system called Jampacker, along with an offline 3D bin packing algorithm (Jampack) that achieves higher packing efficiency.

## Multiple-bin-type 3D bin packing problems in open sizes

For the complex challenge of open-dimensional 3D bin packing involving multiple-bin types, *Tsai & Li (2006)* proposed a highly effective global optimization method. Their approach converts the 3D-ODRPP (3D open-dimensional rectangular items problem) into a hybrid integer linear programming problem, enabling efficient solutions to the problem. Expanding on this research, *Tsai, Wang & Lin (2015)* focused on the rectangular items problem, which entails placing a given number and size of rectangular items inside a large rectangle with the minimum possible area. Another notable contribution in this field was made by *Junqueira & Morabito (2017)*, who introduced the 3D open-dimensional rectangular packing problem. The objective of this problem is to determine the minimum volume of a rectangular container needed to pack a set of rectangular items. To solve this problem, an optimization solver, location-free hybrid-integer programming (MIP) formulation, and a segmented linearization method were employed. Furthermore, *Lin, Tsai & Chang (2017)* investigated the 3D-ODRPP. Their innovative approach involves transforming the original problem into a hybrid-integer linear programming problem. This reformulation reduces the number of constraints by utilizing discrete variables to linearize symbolic terms.

In summary, the current research primarily focuses on single-bin size variable and non-variable problems, as well as multi-bin size non-variable problems. For the fixed-size multi-bin 3D bin packing problem, researchers have proposed various heuristic algorithms and mathematical models to find optimal bin packing solutions. Similarly, for the open-size single-bin 3D bin packing problem, different algorithms and models have been employed to address different variations and achieve goals such as reducing packing volume and creating economic and ecological benefits. Additionally, for the problem of multiple-bin types with fixed dimensions, researchers have proposed diverse heuristic algorithms and mathematical models to find optimal bin-packing solutions.

However, the research on 3D packing problems with multiple bin types in the field of e-commerce warehousing has been more limited so far, and the existing research suffers from the problems of high model complexity and long computation time, although some models and methods have been proposed to solve the 3D packing problem with a single bin type in the previous researches. However, facing the open-size 3D packing problem (3D-MODRPP) with many different bin types, the existing methods have high complexity and long computation time, which makes it difficult to be effectively solved in practical applications. This leads to the problem of inefficiency and excessive computational cost in

**Table 1  Comparison between the study of open size 3D bin packing problem.**

| Author | Research Questions | Research methods and shortcomings |
|---|---|---|
| *Litvinchev, Pankratov & Romanova (2019)* | Open size single bin type 3D bin packing problem | The problem is formulated using the phi function technique, which utilizes an analytical description of the relationship between irregular objects. The modeling approach involves formulating the problem as a nonlinear programming problem. However, it should be noted that the assumption made in the model, where each object is approximated by a polyhedron, may not precisely capture the exact shape of the real objects involved. |
| *Que, Yang & Zhang (2023)* | Open size single bin type 3D bin packing problem | The DRL model utilizes the Transformer architecture as a policy network and is trained through a neighborhood policy optimization algorithm. However, there are certain drawbacks associated with DRL. Firstly, DRL necessitates a substantial amount of problem instances for effective training. Secondly, the performance of the model is influenced by both the quality and quantity of data available for training. |
| *Tsai, Wang & Lin (2015)* | 3D bin packing problem for open size multiple-bin types | The 3D-ODRPP is converted into a hybrid integer linear programming problem to effectively address the challenge of finding optimal solutions for the 3D open-dimensional rectangular packing problem. However, it should be noted that the proposed model exhibits a high level of complexity, leading to long computation times. |
| This article | 3D bin packing problem for open size multiple-bin types | A 3D-MODRPP hybrid integer planning model is developed that allows for the existence of multiple open dimensions with different lengths, widths, and height dimension fetches based on the 3D-ODRPP problem *Tsai, Wang & Lin (2015)*. A new 0–1 integer variable assumption method is proposed to reduce the model complexity and computation time. |

dealing with multiple bin types and open-size scenarios in logistics enterprises such as the e-commerce warehousing field, despite the existence of some mature solutions.

Therefore, in this article, by constructing a mixed integer planning model and adopting a new 0–1 integer variable assumption method, we conduct an in-depth study on the open-size 3D packing problem for multiple bin types in the e-commerce context on the basis of existing research. This study successfully reduces the number of constraints, which effectively reduces the complexity of the model and accelerates the computational process. It also achieves more excellent results based on a dataset that is closer to the 3D-MODRPP problem. Table 1 illustrates a comparison between the existing research on the open 3D bin packing problem and the proposed study on the open-size multiple-bin type bin packing problem.

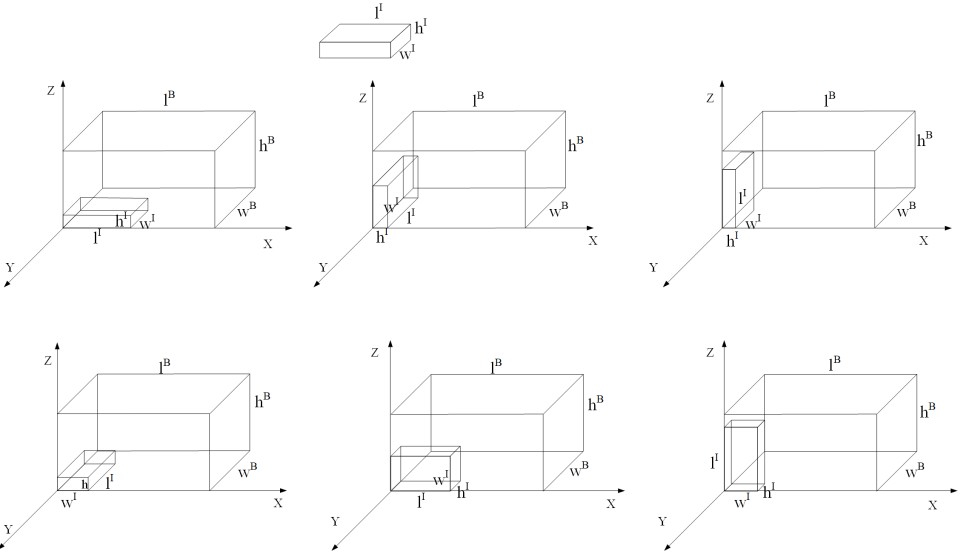

**Figure 1** Six kinds of rectangular items rotation direction.

## Problem description

In this study, we investigate the 3D-MODRPP, which involves arranging a batch of rectangular items into various packing bins categorized by their dimensions. The 3D-MODRPP specifically focuses on the organization of rectangular items within a designated bin, considering different bin types based on their dimensions. The application scenario for this problem lies within the realm of e-commerce warehouse packing, where multiple-bin sizes are typically available. When processing different sets of orders, the packing operator must select the appropriate bin to accommodate all the items in the order, aiming to maximize space utilization. Our optimization objective in this study is to achieve the highest loading rate for the packing bin. We allow the rotation of items in six different directions, as illustrated in Fig. 1. Where, $l^B, w^B, h^B$ denote the length, width, and height of the packing bin respectively, and $l^I, w^I, h^I$ denote the length, width, and height of the item respectively. It is worth noting that in the e-commerce storage bin packing process, fillers are commonly utilized to secure or support the items. However, this research does not focus on the investigation of items' support conditions. The constraints considered in this article's research problem are as follows:

(1) Each shipment requires considering six different rotation directions for the bin.
(2) It is essential to ensure that the bin remains within the boundaries of the packing bin and does not exceed them.
(3) To maintain proper packing, it is necessary to prevent any overlap between the items inside the packing bin.

# VARIABLE ASSUMPTIONS AND MODEL CONSTRUCTION

## Variable assumptions

Variables can be categorized into two types: conditional variables and decision variables. Conditional variables are those that are known or determined in advance, while decision variables are those that require experimental calculations or decision-making processes.

(1) Conditional variables

$n$: Indicates the number of bin type categories;

$q$: Indicates the number of orders;

$m_r$: Indicates the number of items in the first order, where $r = 1, 2, q$;

$l_{i_r}^I, w_{i_r}^I, h_{i_r}^I$: Denote the length with, and height of the first item in the first order, respectively, where $i_r = 1, 2, \ldots, m_r$;

$l_{i_r}^T, w_{i_r}^T, h_{i_r}^T$: Denotes the dimensions values in the axes, axes, and axes dimensions after rotation of the first item in the first order into the bin, *i.e.,* the length, width, and height of them after rotation, respectively, where $i_r = 1, 2, \ldots, m_r$;

$M$: Denotes a sufficiently large positive number.

(2) Decision variables

$x_{i_r}, y_{i_r}, z_{i_r}$: The coordinates of the packing position of the first item in the axes, axes, and axes dimensions, respectively, using the lower left and right corners of the bin into which the first order is loaded as the origin and establishing a three-dimensional right-angle coordinate system;

$l_j^B, w_j^B, h_j^B$: Denote the length, width, and height of the first type of bin, respectively, $j = 1, 2, \ldots, n$; $R_{r,j}$: Indicates whether all items in the first order are loaded into the first bin, and is a 0–1 variable, where 1 means loaded and 0 means not loaded, where $r = 1, 2, \ldots, q, j = 1, 2, \ldots, n$;

$T_{11}^{i_r}, T_{12}^{i_r}, T_{13}^{i_r}, T_{21}^{i_r}, T_{22}^{i_r}, T_{23}^{i_r}, T_{31}^{i_r}, T_{32}^{i_r}, T_{33}^{i_r}$: Indicates the direction of rotation of the item, as a 0–1 variable;

$D_{i_r, i_r'}^k$: A 0–1 variable indicating whether two items overlap with each other, $, I_r \neq i_r', I_r, I_r' = 1, 2, \ldots, m, k = 1, 2, 3, 4, 5, 6$, The equation determines whether every two items overlap by using six 0–1 variables;

$O_{i_r, i_r'}^1, O_{i_r, i_r'}^2, O_{i_r, i_r'}^3$: A 0–1 variable is used to indicate whether there is an overlap between two items. To determine whether each pair of items overlaps, only three 0–1 variables are needed; $U_1^{i_r}, U_2^{i_r}, U_3^{i_r}, U_4^{i_r}, U_3^{i_r}, U_6^{i_r}$: A 0–1 variable representing the six directions of item rotation.

## Model construction

(1) The objective function of the model is to maximize the volume of all items within various types of bines, aiming to achieve the highest loading rate.

$$max \quad z = \frac{\sum_{r=1}^{q} \sum_{i_r=1}^{m_r} l_{i_r}^I \cdot w_{i_r}^I \cdot h_{i_r}^I}{\sum_{r=1}^{q} \sum_{j=1}^{n} R_{r,j} \cdot l_j^B \cdot w_j^B \cdot h_j^B} \tag{1}$$

(2) Constraints

(a) Rotation constraints

Assume that the item can be rotated in 6 directions.

$$l_{i_r}^T = T_{11}^{i_r} \cdot l_{i_r}^I + T_{12}^{i_r} \cdot w_{i_r}^I + T_{13}^{i_r} \cdot h_{i_r}^I \tag{2}$$

$$w_{i_r}^T = T_{21}^{i_r} \cdot l_{i_r}^I + T_{22}^{i_r} \cdot w_{i_r}^I + T_{23}^{i_r} \cdot h_{i_r}^I \tag{3}$$

$$h_{i_r}^T = T_{31}^{i_r} \cdot l_{i_r}^I + T_{32}^{i_r} \cdot w_{i_r}^I + T_{33}^{i_r} \cdot h_{i_r}^I \tag{4}$$

$$T_{11}^{i_r} + T_{12}^{i_r} + T_{13}^{i_r} = 1 \tag{5}$$

$$T_{21}^{i_r} + T_{22}^{i_r} + T_{23}^{i_r} = 1 \tag{6}$$

$$T_{31}^{i_r} + T_{32}^{i_r} + T_{33}^{i_r} = 1 \tag{7}$$

$$T_{11}^{i_r} + T_{21}^{i_r} + T_{31}^{i_r} = 1 \tag{8}$$

$$T_{12}^{i_r} + T_{22}^{i_r} + T_{32}^{i_r} = 1 \tag{9}$$

$$T_{13}^{i_r} + T_{23}^{i_r} + T_{33}^{i_r} = 1 \tag{10}$$

The rotation constraints for items described in Eqs. (2) to (10) require representing the rotation direction of each item using nine 0–1 variables and nine constraints. It is possible to simplify the model by reducing the number of 0–1 variables and constraints, as outlined below.

$$l_{i_r}^T = U_1^{i_r} \cdot l_{i_r}^I + U_2^{i_r} \cdot l_{i_r}^I + U_3^{i_r} \cdot w_{i_r}^I + U_4^{i_r} \cdot w_{i_r}^I + U_5^{i_r} \cdot h_{i_r}^I + U_6^{i_r} \cdot h_{i_r}^I \tag{2'}$$

$$w_{i_r}^T = U_1^{i_r} \cdot w_{i_r}^I + U_2^{i_r} \cdot h_{i_r}^I + U_3^{i_r} \cdot l_{i_r}^I + U_4^{i_r} \cdot h_{i_r}^I + U_5^{i_r} \cdot l_{i_r}^I + U_6^{i_r} \cdot w_{i_r}^I \tag{3'}$$

$$h_{i_r}^T = U_1^{i_r} \cdot h_{i_r}^I + U_2^{i_r} \cdot w_{i_r}^I + U_3^{i_r} \cdot h_{i_r}^I + U_4^{i_r} \cdot l_{i_r}^I + U_{i_r}^{r} \cdot W_{i_r}^{i_r} \cdot w_{i_r}^I \cdot w_{i_r}^I + U_6^{i_r} \cdot l_{\dot{r}}^I \tag{4'}$$

$$U_1^{i_r} + U_2^{i_r} + U_3^{i_r} + U_4^{i_r} + U_5^{i_r} + U_6^{i_r} = 1 \tag{5'}$$

(b) Transboundary constraints

The cross-boundary constraint ensures that the boxed items cannot extend beyond the boundaries of the packing bin, and all items must remain inside the packing bin.

$$x_{i_r} + l_{i_r}^T \le \sum_{j=1}^{n} R_{r,j} \cdot l_j^B \tag{11}$$

$$y_{i_r} + w_{i_r}^T \le \sum_{j=1}^{n} R_{r,j} \cdot w_j^B \tag{12}$$

$$z_{i_r} + h_{i_r}^T \le \sum_{j=1}^{n} R_{r,j} \cdot h_j^B \tag{13}$$

After substituting Eqs. (2) into (11) and (3) into (12) and (4) into (13), respectively, the following equation can be obtained:

$$x_{i_r} + U_1^{i_r} \cdot l_{i_r}^I + U_2^{i_r} \cdot l_{i_r}^I + U_3^{i_r} \cdot w_{i_r}^I + U_4^{i_r} \cdot w_{i_r}^I + U_5^{i_r} \cdot h_{i_r}^I + U_6^{i_r} \cdot h_{i_r}^I \le \sum_{j=1}^{n} R_{r,j} \cdot l_j^B \tag{11'}$$

$$y_{i_r} + U_1^{i_r} \cdot w_{i_r}^I + U_2^{i_r} \cdot h_{i_r}^I + U_3^{i_r} \cdot l_{i_r}^I + U_4^{i_r} \cdot h_{i_r}^I + U_5^{i_r} \cdot l_{i_r}^I + U_6^{i_r} \cdot w_{i_r}^I \leq \sum_{j=1}^{n} R_{r,j} \cdot w_j^B \tag{12'}$$

$$z_{i_r} + U_1^{i_r} \cdot h_{i_r}^I + U_2^{i_r} \cdot w_{i_r}^I + U_3^{i_r} \cdot h_{i_r}^I + U_4^{i_r} \cdot l_{i_r}^I + U_5^{i_r} \cdot w_{i_r}^I + U_6^{i_r} \cdot l_{i_r}^I \leq \sum_{j=1}^{n} R_{r,j} \cdot h_j^B \tag{13'}$$

In conjunction with Eqs. (5) to (10), these equations represent the non-transgression constraint following item rotation.

(c) Overlapping constraints

Overlap constraint means that two items within the same order cannot overlap in spatial extent.

$$x_{i'_r} + l_{i'_r}^T \leq x_{i_r} + (1 - D_{i_r, i'_r}^1) \cdot M \tag{14}$$

$$x_{i_r} + l_{i_r}^T \leq x_{i'_r} + (1 - D_{i_r, i'_r}^2) \cdot M \tag{15}$$

$$y_{i'_r} + w_{i'_r}^T \leq y_{i_r} + (1 - D_{i_r, i'_r}^3) \cdot M \tag{16}$$

$$y_{i_r} + w_{i_r}^T \leq y_{i'_r} + (1 - D_{i_r, i'_r}^4) \cdot M \tag{17}$$

$$z_{i'_r} + h_{i'_r}^T \leq z_{i_r} + (1 - D_{i_r, i'_r}^5) \cdot M \tag{18}$$

$$z_{i_r} + h_{i_r}^T \leq z_{i'_r} + (1 - D_{i_r, i'_r}^6) \cdot M \tag{19}$$

$$D_{i_r, i'_r}^1 + D_{i_r, i'_r}^2 + D_{i_r, i'_r}^3 + D_{i_r, i'_r}^4 + D_{i_r, i'_r}^5 + D_{i_r, i'_r}^6 \geq 1 \tag{20}$$

According to Eqs. (2) to (4), the length, width, and height dimensions of the rotated items in Eqs. (14) to (20) are obtained by substituting.

$$x_{i'_r} + U_1^{i'_r} \cdot l_{i'_r}^I + U_2^{i'_r} \cdot l_{i'_r}^I + U_3^{i'_r} \cdot w_{i'_r}^I + U_4^{i'_r} \cdot w_{i'_r}^I + U_5^{i'_r} \cdot h_{i'_r}^I + U_6^{i'_r} \cdot h_{i'_r}^I \leq x_{i_r} + (1 - D_{i_r, i'_r}^1) \cdot M$$
$$\tag{14'}$$

$$x_{i_r} + U_1^{i_r} \cdot l_{i_r}^I + U_2^{i_r} \cdot l_{i_r}^I + U_3^{i_r} \cdot w_{i_r}^I + U_4^{i_r} \cdot w_{i_r}^I + U_5^{i_r} \cdot h_{i_r}^I + U_6^{i_r} \cdot h_{i_r}^I \leq x_{i'_r} + (1 - D_{i_r, i'_r}^2) \cdot M$$
$$\tag{15'}$$

$$y_{i'_r} + U_1^{i'_r} \cdot w_{i'_r}^I + U_2^{i'_r} \cdot h_{i'_r}^I + U_3^{i'_r} \cdot l_{i'_r}^I + U_4^{i'_r} \cdot h_{i'_r}^I + U_5^{i'_r} \cdot l_{i'_r}^I + U_6^{i'_r} \cdot w_{i'_r}^I \leq y_{i_r} + (1 - D_{i_r, i'_r}^3) \cdot M$$
$$\tag{16'}$$

$$y_{i_r} + U_1^{i_r} \cdot w_{i_r}^I + U_2^{i_r} \cdot h_{i_r}^I + U_3^{i_r} \cdot l_{i_r}^I + U_4^{i_r} \cdot h_{i_r}^I + U_5^{i_r} \cdot l_{i_r}^I + U_6^{i_r} \cdot w_{i_r}^I \leq y_{i'_r} + (1 - D_{i_r, i'_r}^4) \cdot M$$
$$\tag{17'}$$

$$z_{i'_r} + U_1^{i'_r} \cdot h_{i'_r}^I + U_2^{i'_r} \cdot w_{i'_r}^I + U_3^{i'_r} \cdot h_{i'_r}^I + U_4^{i'_r} \cdot l_{i'_r}^I + U_5^{i'_r} \cdot w_{i'_r}^I + U_6^{i'_r} \cdot l_{i'_r}^I \leq z_{i_r} + (1 - D_{i_r, i'_r}^5) \cdot M$$
$$\tag{18'}$$

$$z_{i_r} + U_1^{i_r} \cdot h_{i_r}^I + U_2^{i_r} \cdot w_{i_r}^I + U_3^{i_r} \cdot h_{i_r}^I + U_4^{i_r} \cdot l_{i_r}^I + U_5^{i_r} \cdot w_{i_r}^I + U_6^{i_r} \cdot l_{i_r}^I \le z_{i'_r} + (1 - D_{i_r, i'_r}^6) \cdot M$$

(19')

Equations (14') to (19') and (20) collectively establish a non-overlapping constraint between items when packing the same order of items into a single bin. *Tsai, Wang & Lin (2015)* proposed a method to reduce the number of 0–1 decision variables when representing the non-overlapping constraints between two items. Instead of using six 0–1 variables, Tsai's approach utilizes only three 0–1 variables. Although Tsai's study focuses on the 3D-ODRPP problem, which differs slightly from the problem addressed in this article, we can adapt (*Tsai, Wang & Lin, 2015*)'s research ideas to reduce the non-overlapping constraint in this article. Consequently, Eqs. (14') to (19') and (20') are modified as follows:

$$x_{i'_r} + U_1^{i'_r} \cdot l_{i'_r}^I + U_2^{i'_r} \cdot l_{i'_r}^I + U_3^{i'_r} \cdot w_{i'_r}^I + U_4^{i'_r} \cdot w_{i'_r}^I + U_5^{i'_r} \cdot h_{i'_r}^I + U_6^{i'_r} \cdot h_{i'_r}^I \le$$
$$x_{i_r} + O_{i_r, i'_r}^1 + O_{i_r, i'_r}^2 + (1 - O_{i_r, i'_r}^3) \cdot M$$

(14")

$$x_{i_r} + U_1^{i_r} \cdot l_{i_r}^I + U_2^{i_r} \cdot l_{i_r}^I + U_3^{i_r} \cdot w_{i_r}^I + U_4^{i_r} \cdot w_{i_r}^I + U_5^{i_r} \cdot h_{i_r}^I + U_6^{i_r} \cdot h_{i_r}^I$$
$$\le x_{i'_r} + O_{i_r, i'_r}^1 + (1 - O_{i_r, i'_r}^2) \cdot M + O_{i_r, i'_r}^3$$

(15")

$$y_{i'_r} + U_1^{i'_r} \cdot w_{i'_r}^I + U_2^{i'_r} \cdot h_{i'_r}^I + U_3^{i'_r} \cdot l_{i'_r}^I + U_4^{i'_r} \cdot h_{i'_r}^I + U_5^{i'_r} \cdot l_{i'_r}^I + U_6^{i'_r} \cdot w_{i'_r}^I$$
$$\le y_{i_r} + (1 - O_{i_r, i'_r}^1) \cdot M + O_{i_r, i'_r}^2 + O_{i_r, i'_r}^3$$

(16")

$$y_{i_r} + U_1^{i_r} \cdot w_{i_r}^I + U_2^{i_r} \cdot h_{i_r}^I + U_3^{i_r} \cdot l_{i_r}^I + U_4^{i_r} \cdot h_{i_r}^I + U_5^{i_r} \cdot l_{i_r}^I + U_6^{i_r} \cdot w_{i_r}^I$$
$$\le y_{i'_r} + O_{i_r, i'_r}^1 + (1 - O_{i_r, i'_r}^2) \cdot M + (1 - O_{i_r, i'_r}^3) \cdot M$$

(17")

$$z_{i'_r} + U_1^{i'_r} \cdot h_{i'_r}^I + U_2^{i'_r} \cdot w_{i'_r}^I + U_3^{i'_r} \cdot h_{i'_r}^I + U_4^{i'_r} \cdot l_{i'_r}^I + U_5^{i'_r} \cdot w_{i'_r}^I + U_6^{i'_r} \cdot l_{i'_r}^I$$
$$\le z_{i_r} + (1 - O_{i_r, i'_r}^1) \cdot M + O_{i_r, i'_r}^2 + (1 - O_{i_r, i'_r}^3) \cdot M$$

(18")

$$z_{i_r} + U_1^{i_r} \cdot h_{i_r}^I + U_2^{i_r} \cdot w_{i_r}^I + U_3^{i_r} \cdot h_{i_r}^I + U_4^{i_r} \cdot l_{i_r}^I + U_5^{i_r} \cdot w_{i_r}^I + U_6^{i_r} \cdot l_{i_r}^I$$
$$\le z_{i'_r} + (1 - O_{i_r, i'_r}^1) \cdot M + (1 - O_{i_r, i'_r}^2) \cdot M + O_{i_r, i'_r}^3$$

(19")

$$O_{i_r, i'_r}^1 + O_{i_r, i'_r}^2 + O_{i_r, i'_r}^3 \ge 1$$

(20')

$$O_{i_r, i'_r}^1 + O_{i_r, i'_r}^2 + O_{i_r, i'_r}^3 \le 2$$

(20")

As depicted in Eqs. (21) to (24), certain variables possess specific values, while others are subject to constraints:

$$x_{i_r}, y_{i_r}, z_{i_r} l_j^B, w_j^B, h_j^B \ge 0$$

(21)

$$R_{r,j}, D_{i_r, i'_r}^k, O_{i_r, i'_r}^1, O_{i_r, i'_r}^2, O_{i_r, i'_r}^3, U_1^{i_r}, U_2^{i_r}, U_3^{i_r}, U_4^{i_r}, U_5^{i_r}, U_6^{i_r} = 0 \vee 1$$

(22)

$$j = 1, 2, \ldots, n; r = 1, 2, \ldots, q$$

(23)

**Table 2** SKU item size information.

| Item size | SKU | | | | | | | | | | | | | | | |
|---|---|---|---|---|---|---|---|---|---|---|---|---|---|---|---|---|
| | 01 | 02 | 03 | 04 | 05 | 06 | 07 | 08 | 09 | 10 | 11 | 12 | 13 | 14 | 15 | 16 |
| Lenght | 25 | 20 | 16 | 15 | 22 | 20 | 2 | 3 | 4 | 5 | 56 | 57 | 39 | 51 | 61 | 67 |
| Width | 8 | 10 | 7 | 12 | 8 | 10 | 2 | 3 | 4 | 5 | 50 | 43 | 34 | 19 | 45 | 48 |
| Height | 6 | 5 | 3 | 6 | 3 | 4 | 2 | 3 | 4 | 5 | 29 | 25 | 30 | 4 | 25 | 27 |

$$i_r \neq i'_r, i_r, i'_r = 1, 2, \ldots, m; k = 1, 2, 3, 4, 5, 6 \tag{24}$$

# COMPUTATIONAL EXPERIMENTS AND SYSTEM DESIGN

In order to verify the effectiveness and efficiency of the proposed model and method, this article conducts a series of computational experiments using arithmetic examples extracted from relevant literature and actual production data. These bins are divided into two categories: The first category consists of item SKU data (Stock Keeping Unit), encompassing 16 different item sizes. The second category involves order data, which aims to validate the applicability of the proposed method across various scenarios and compare it with related models. The computational equipment used in this study includes an Intel(R) Core(TM) i7-10510U CPU @ 1.80 GHz, 16.0 GB of RAM (15.8 GB available), and the Gruobi10.01 solver.

## Example data

The information regarding all item SKU sizes is presented in Table 2, encompassing a total of 16 distinct sizes. These sizes have been derived using the arithmetic method of *Tsai, Wang & Lin (2015)*.

Ten different problem arithmetic bins were developed based on the types of items, and each problem arithmetic bin was loaded into a packing bin model. Table 3 provides detailed information on the type and number of items present in each arithmetic bin. Questions 1 through 4 represent scenarios involving strongly heterogeneous items, while Questions 5 and 6 are for weakly heterogeneous items. Questions 7 through 10 are derived from actual production data obtained from different firms. The arithmetic examples are from *Tsai, Wang & Lin (2015)*.

## Algorithm solving

(1) Strong heterogeneous packing problem

Problems 1 to 4 fall under the category of strongly heterogeneous item packing problems, where "strongly heterogeneous" implies that there are very few items with identical shapes and sizes. In this study, we employ a hybrid integer programming model and utilize the GUROBI solver to compute and compare the results with the approach presented by Tsai and *Li, Chen & Huo (2022)*, as well as *Tsai, Wang & Lin (2015)*. The obtained results are summarized in Table 4.

As shown in Table 4, the computational time of the proposed model in this article is significantly reduced compared to the CPU computation time while achieving the

**Table 3  Number of SKU items included in the algorithm.**

| Examples | SKU | | | | | | | | | | | | | | | |
|---|---|---|---|---|---|---|---|---|---|---|---|---|---|---|---|---|
| | 01 | 02 | 03 | 04 | 05 | 06 | 07 | 08 | 09 | 10 | 11 | 12 | 13 | 14 | 15 | 16 |
| Problem01 | 1 | 1 | 1 | 1 | | | | | | | | | | | | |
| Problem02 | 1 | 1 | 1 | 1 | 1 | | | | | | | | | | | |
| Problem03 | 1 | 1 | 1 | 1 | 1 | 1 | | | | | | | | | | |
| Problem04 | 2 | 1 | 1 | 1 | 1 | 1 | | | | | | | | | | |
| Problem05 | | | | | | | 3 | 3 | 1 | 1 | | | | | | |
| Problem06 | | | | | | | 3 | 3 | 2 | 1 | | | | | | |
| Problem07 | | | | | | | | | | | 1 | 1 | 1 | 1 | | |
| Problem08 | | | | | | | | | | | 1 | 1 | 1 | 1 | 1 | |
| Problem09 | | | | | | | | | | | 2 | 1 | 1 | 1 | 1 | |
| Problem10 | | | | | | | | | | | 2 | 1 | 1 | 1 | 1 | 1 |

**Table 4  Results of the three methods for solving the strongly heterogeneous packing problem.**

| Examples | Item | *Tsai & Li (2006)* | *Tsai, Wang & Lin (2015)* | Proposed method |
|---|---|---|---|---|
| Problem01 | CPU time (s) | 1,758 | 245 | 0.44 |
| | No. of total 0–1 variables | 765 | 75 | 43 |
| | No. of total constraints | 5,376 | 129 | 68 |
| | Solution (L, W, H) | (28, 26, 6) | (28, 26, 6) | (28, 26, 6) |
| | Objective value: (L, W, H) | 4,368 | 4,368 | 4,368 |
| Problem02 | CPU time (s) | 13,123 | 520 | 2.35 |
| | No. of total 0–1 variables | 840 | 96 | 60 |
| | No. of total constraints | 5,491 | 166 | 100 |
| | Solution (L, W, H) | (30, 28, 6) | (30, 28, 6) | (30, 28, 6) |
| | Objective value: (L, W, H) | 5,040 | 5,040 | 5,040 |
| Problem03 | CPU time (s) | 84,588 | 20,641 | 7.57 |
| | No. of total 0–1 variables | 864 | 120 | 81 |
| | No. of total constraints | 5,535 | 210 | 144 |
| | Solution (L, W, H) | (35, 28, 6) | (35, 28, 6) | (35, 28, 6) |
| | Objective value: LWH | 5,880 | 5,880 | 5,880 |
| Problem04 | CPU time (s) | >86,400 | 46,578 | 20.69 |
| | No. of total 0–1 variables | 891 | 147 | 105 |
| | No. of total constraints | 5,586 | 261 | 196 |
| | Solution (L, W, H) | – | (40, 22, 14) | (40, 22, 14) |
| | Objective value: (L, W, H) | – | 12,320 | 12,320 |

same experimental results. Additionally, the present model reduces the number of 0–1 variables in both overlap and rotation constraints. In the original *Tsai, Wang & Lin (2015)* method, problem 1 was reduced from 75 to 43 variables, problem 2 from 96 to 60 variables, problem 3 from 120 to 81 variables, and problem 4 from 147 to 105 variables. Consequently, the number of constraints is also reduced. For instance, problem 3, which consists of six items with different SKU types, saw a decrease in CPU

**Table 5  Results of the three methods for solving the weakly heterogeneous classification and packing problem.**

| Examples | Item | Tsai & Li (2006) | Tsai, Wang & Lin (2015) | Proposed method |
|---|---|---|---|---|
| Problem05 | CPU time (s) | 75 | 9 | 0.52 |
| | No. of total 0–1 variables | 240 | 156 | 132 |
| | Solution (L, W, H) | (9, 8, 5) | (9, 8, 5) | (9, 8, 5) |
| Problem06 | CPU time (s) | 186 | 43 | 1.25 |
| | No. of total 0–1 variables | 297 | 180 | 162 |
| | Solution (L, W, H) | (10, 8, 6) | (10, 8, 6) | (10, 8, 6) |

computation time from 20,641 s in *Tsai, Wang & Lin (2015)* method to 20.69 s in the proposed method. This reduction in variables simplifies the model's complexity and enhances its ability to solve the highly heterogeneous bin packing problem.

(2)   Weak heterogeneous packing problem
    Problems 5 and 6 are categorized as weakly heterogeneous item bin packing problems, where "weakly heterogeneous item bin packing problem" refers to the classification of items into a relatively small number of categories, with each category consisting of identical items in terms of shape and size. Table 5 presents the computational results for these two weakly heterogeneous bin packing problems and compares them with the method proposed by Tsai and *Li, Chen & Huo (2022)*, as well as *Tsai, Wang & Lin (2015)*. As shown in Table 5, our model significantly reduces the utilization of 0–1 variables. Furthermore, in terms of computational time, the proposed method in this article demonstrates the shortest CPU time for solving the model. The complexity of solving these problems is also lower compared to solving the strongly heterogeneous classification packing problem, mainly due to the reduced need for combinatorial strategies when rotating the bins. The computational results indicate that the proposed method effectively addresses the challenges posed by the weakly heterogeneous packing problem.

(3)   Actual bin packing problem
    Problems 7 to 10 in this article are based on the packing example of the enterprise and were solved using the GUROBI solver. GUROBI is a highly efficient mathematical optimization solver used for solving various optimization problems, including linear programming, integer programming, and quadratic programming. It is widely employed in operations research, industrial optimization, logistics planning, and similar fields. In this study, computational results were obtained using GUROBI and compared with those obtained from LINGO and CPLEX. Table 6 presents the comparative findings. GUROBI demonstrates exceptional computational performance on the packing instance data, requiring less CPU time than LINGO and CPLEX to solve the reconfigured hybrid integer planning model. For problem 9, involving five items, GUROBI produces a solution within 5 s, while both LINGO and CPLEX require CPU times exceeding 60 min. As the proposed model in this article is linear and of

**Table 6 Results of the three methods for solving the weakly heterogeneous classification and packing problem.**

| Examples | Item | *Tsai, Wang & Lin (2015)* CPU time by LINGO | *Tsai, Wang & Lin (2015)* CPU time by CPLEX | Proposed method CPU time by GUROBI |
|---|---|---|---|---|
| Problem07 | Solution (L, W, H) | (127, 57, 30) | (127, 57, 30) | (127, 57, 30) |
|  | CPU time (s) | 66 | 2 | 0.50 |
| Problem08 | Solution (L, W, H) | (102, 95, 30) | (102, 95, 30) | (102, 95, 30) |
|  | CPU time (s) | 640 | 7 | 1.54 |
| Problem09 | Solution (L, W, H) | (92, 81, 50) | (92, 81, 50) | (92, 81, 50) |
|  | CPU time (s) | 01:30:42 | 00:01:16 | 4.38 |
| Problem10 | Solution (L, W, H) | (92, 81, 50) | (92, 81, 50) | (92, 81, 50) |
|  | CPU time (s) | 49503 | 414 | 135.31 |

low complexity, the computation time is significantly lower compared to the nonlinear model employed by LINGO.

(4) Bin packing problems for multiple-bin open sizes

The 3D bin packing problem with multiple bin sizes focuses on the application of strong heterogeneous bin packing. Table 7 displays eight randomly generated orders, each comprising two to four items.

The calculation results for the corresponding orders were obtained under various designs of open sizes for different bin types, as presented in Table 8. Orders 01–04 were tested with 2 and 3 bin models, respectively, achieving an average space utilization of 80.16% and 91.92%, respectively, the CPU calculation time remained within 20 s for both orders 01–04. Visual representations of the results for orders 01–04 are displayed, with Fig. 2 showcasing the results for the 2-bin type and Fig. 3 illustrating the results for the 3-bin type.

Orders 01–06 underwent experimentation with packing bin types 2, 3, and 4 individually. Notably, the average space utilization achieved impressive rates of 62.72%, 81.59%, and 90.93% for each bin type, respectively. Moreover, the CPU computation time remained well within the 15-minute limit. Regarding the visualization of the results for orders 01–06, Fig. 4 displays the outcomes obtained using bin type 2, Fig. 5 exhibits the results with bin type 3, and Fig. 6 illustrates the results with bin type 4.

The utilization and performance of orders 01–08 were evaluated using different types of bins: two, three, and four. The average space utilization achieved was 62.24%, 80.43%, and 89.35% respectively. As expected, the loading efficiency improved with an increasing number of bin types. However, it should be noted that the CPU computation time also increased accordingly, staying within 7 min. To visually represent these results, Fig. 7 illustrates the visualization for bin type 2, Fig. 8 for bin type 3, and Fig. 9 for bin type 4. These figures provide a clear depiction of the optimization achieved in terms of loading rate and space utilization as the number of bin types increased, while also indicating the corresponding increase in CPU computation time.

**Table 7  Order item SKU information.**

| Item size | SKU | | | | | | | | | | | | | | | |
|---|---|---|---|---|---|---|---|---|---|---|---|---|---|---|---|---|
| | 01 | 02 | 03 | 04 | 05 | 06 | 07 | 08 | 09 | 10 | 11 | 12 | 13 | 14 | 15 | 16 |
| Order01 | | | | | | | 1 | | | | | | | | | 1 |
| Order02 | | | | 1 | | | | | | | | 1 | | | | |
| Order03 | | | 1 | 1 | | | 1 | | | | | | | | | |
| Order04 | | | | | | | 1 | 1 | | | | | | | | |
| Order05 | | | | | | | | | 1 | | | | | 1 | | |
| Order06 | | 1 | | | | | 1 | 2 | | | | | | | | |
| Order07 | | 1 | | | | | 1 | | 1 | 1 | | | | | | |
| Order08 | | | | | 1 | | | | 1 | 1 | | | | | | |

**Table 8  Results of multiple-bin open size 3D packing problems.**

| Examples | Number of orders | Solution types | Number of (L, W, H) | Space utilisation (%) | CPU time (s) |
|---|---|---|---|---|---|
| Order01–04 | 4 | 2 | (35, 7, 12)<br>(71, 27, 48) | 80.16 | 1.25 |
| | | 3 | (48, 71, 27)<br>(35, 7, 12) | 91.92 | 16.36 |
| | | | (63, 43, 25) | | |
| Order01–06 | 6 | 2 | (35, 7, 12)<br>(67, 31, 48) | 62.73 | 841.51 |
| | | 3 | (48, 71, 27)<br>(35, 7, 12)<br>(43, 34, 30) | 81.59 | 744.90 |
| | | 4 | (27, 71, 48)<br>(35, 7, 12)<br>(63, 43, 25)<br>(43, 33, 30) | 90.93 | 216.10 |
| Order01–08 | 8 | 2 | (67, 31, 48)<br>(26, 20, 6) | 62.24 | 5.45 |
| | | 3 | (71, 48, 27)<br>(26, 20, 6)<br>(34, 43, 30) | 80.43 | 52.94 |
| | | 4 | (71, 27, 48)<br>(26, 20, 6)<br>(43, 30, 34)<br>(63, 43, 25) | 89.35 | 389.87 |

## System design

To effectively integrate the e-commerce warehouse bin packing problem-solving model with the current industry trends of "unmanned" and "intelligent," this study presents a refined design for a fully automatic digital intelligent bin packing system. Leveraging

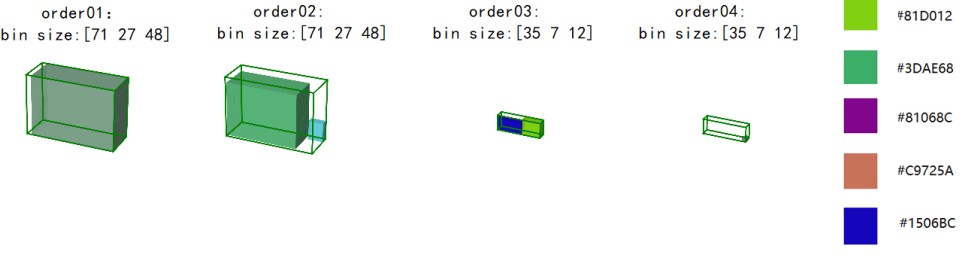

**Figure 2** Visualization of open size 3D packing results for four orders of three bin types.

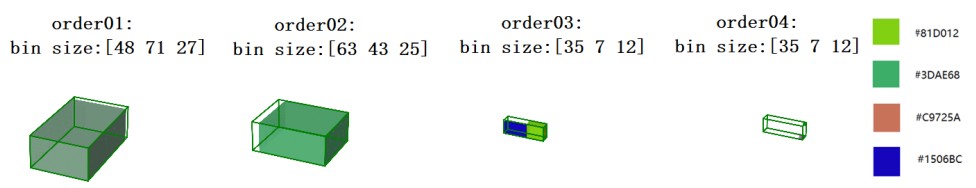

**Figure 3** Visualization of open size 3D packing results for four orders of two bin types.

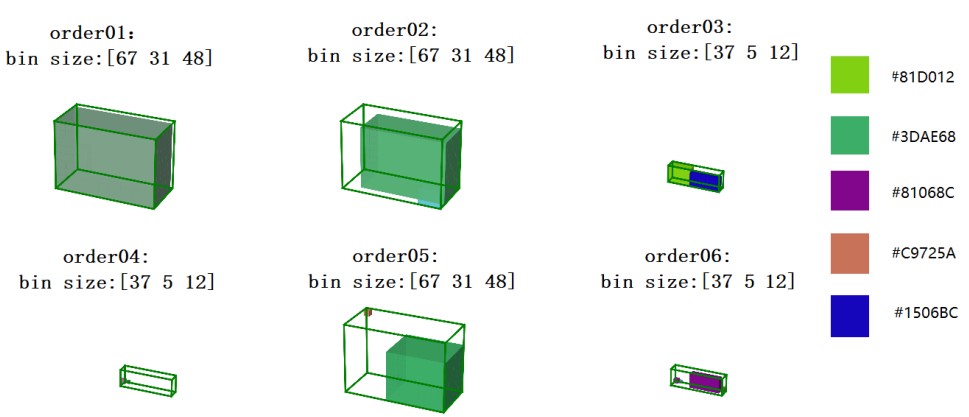

**Figure 4** Visualization of open size 3D packing results for six orders of two bin types.

mature hardware and software, the proposed system aims to enhance operational efficiency and cater to the evolving needs of the industry. Figure 10 illustrates the layout of the fully automatic digital intelligent bin packing system. Comprising three key modules, the digital automatic packing system initiates its operation sequence by inputting order items in module 3. Subsequently, module 2, utilizing the model developed in 'Variable assumptions and model construction' and the algorithm outlined in 'Example data'–'Algorithm solving', adjusts the order and orientation of the packing items. Finally, module 1 completes the packing operation for the order items, concluding the process.

In Fig. 10, it is illustrated that once the picking process is completed, the order items proceed to module 3. Utilizing a chassis equipped with a gravity-sensing device, the system
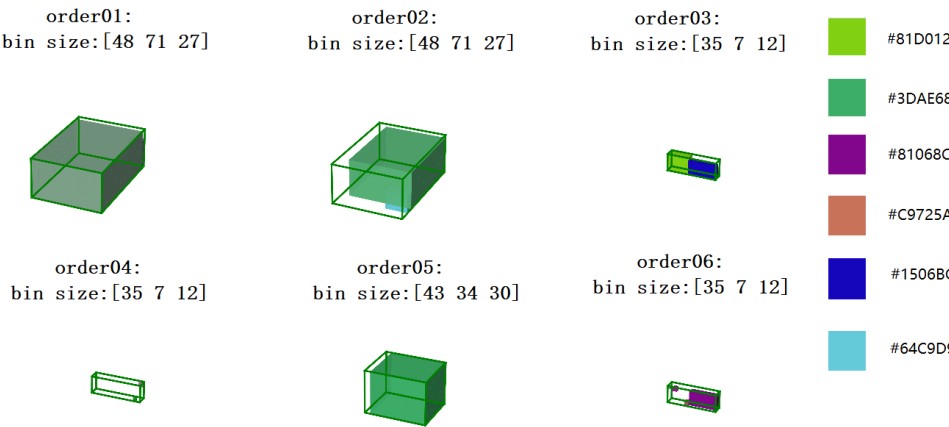

**Figure 5** Visualization of open size 3D packing results for six orders of three bin types.

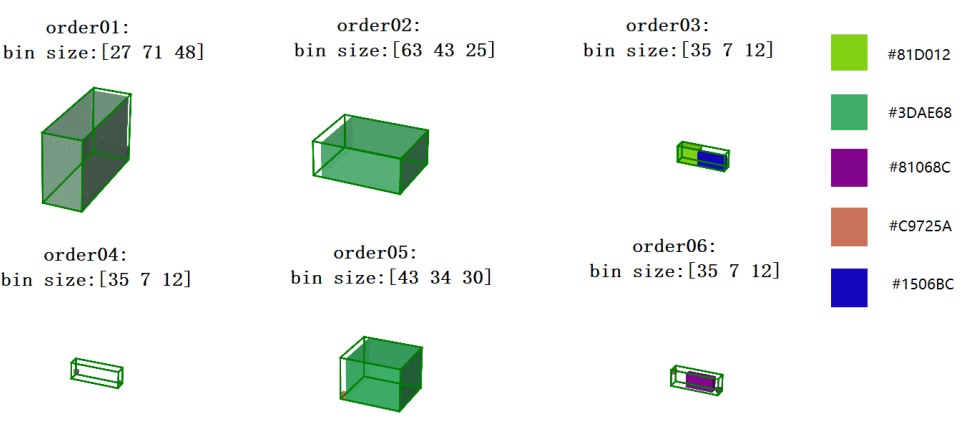

**Figure 6** Visualization of open size 3D packing results for eight orders of two bin types.

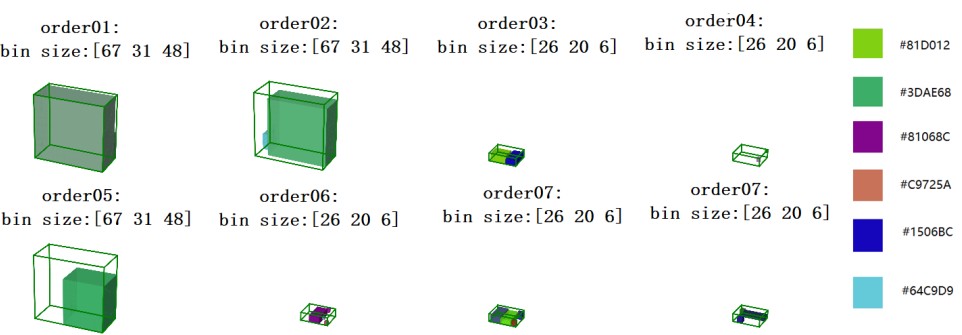

**Figure 7** Visualization of open size 3D packing results for six orders of four bin types.

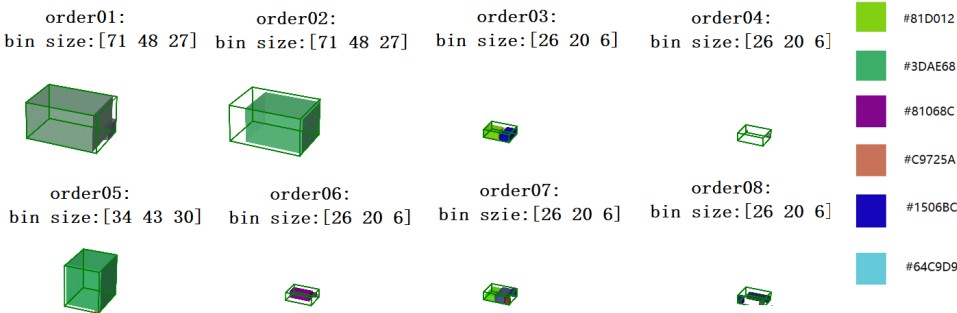

**Figure 8  Visualization of open size 3D packing results for eight orders of three bin types.**

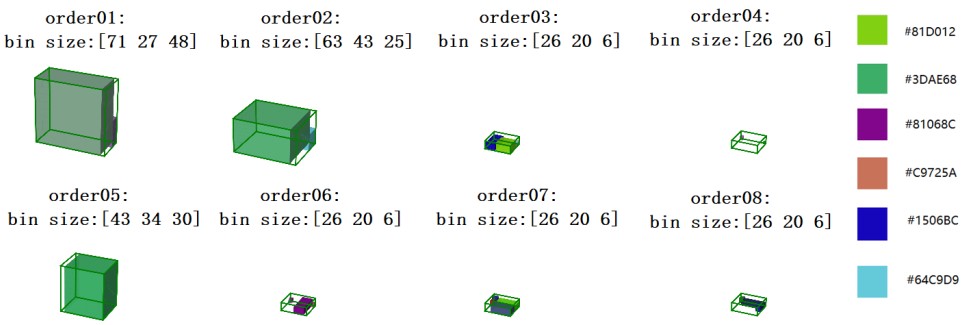

**Figure 9  Visualization of open size 3D packing results for eight orders of four bin types.**

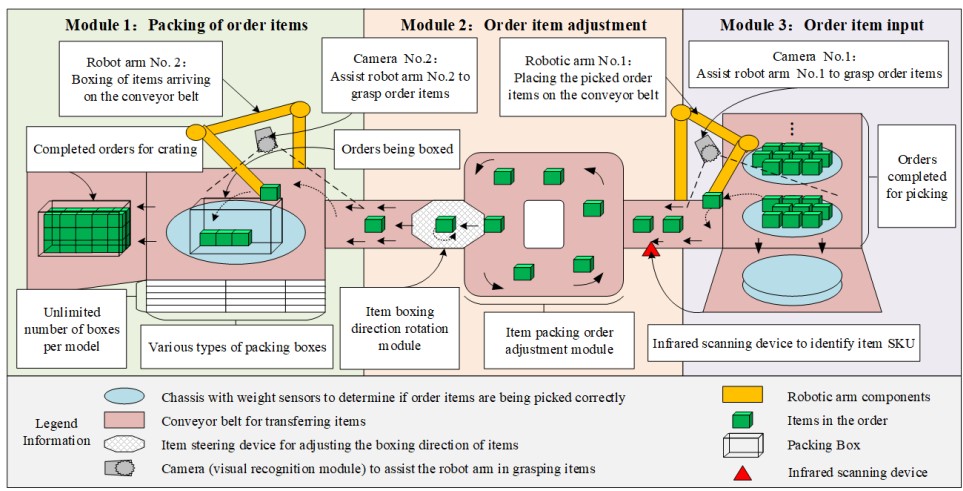

**Figure 10  Schematic diagram of digital automatic bin packing system.**

verifies the alignment between the items positioned on each chassis and the corresponding order information. This step serves as the initial review of the items picked for the order. By combining the efforts of robot arm No. 1 and camera No. 1, the items belonging to the current order are removed from the chassis and sequentially placed onto the conveyor belt in module 2. Consequently, the order items are then directed toward the order and direction adjustment link. The infrared scanning device is positioned next to the conveyor belt in module 2. Its purpose is to scan the items as they pass through in a sequential manner, thereby obtaining the SKU information associated with each item.

Simultaneously, it captures the current sequence of items on the conveyor belt. By comparing this information with the original order details, the device completes the review of the second-order information. In module 2, the intelligent packing strategy calculates a packing scheme. Based on this scheme, the first mechanical arm adjusts the order of items and their placement direction to align with the requirements of the packing scheme. This ensures that the items are properly arranged and oriented before they proceed to module 1 for the packing operation. In module 1, the order items are carefully boxed, and the appropriate type of bin must be selected based on the predetermined packing scheme. The types of bins have been predetermined, and there is an unlimited supply of each type. Spare bines are neatly folded and stored until they are needed, at which point mechanical devices unfold them for use. Within module 1, the second mechanical arm collaborates with camera No. 2 to smoothly and accurately grab the order items from the conveyor belt and place them inside the designated packing bins. Furthermore, the packing bin employed in module 1 is positioned on a chassis equipped with a weight sensor. Once all the items for an order have completed the packing process, the weight-sensing chassis within module 1 can verify the correctness of the packing operation, thereby fulfilling the role of a third review. This comprehensive approach ensures that the entire order-packing operation is completed with utmost precision. Throughout the process, the order information undergoes three rounds of review, guaranteeing a high level of accuracy. Moreover, the packing scheme strictly adheres to the intelligent packing strategy's calculation results, enabling the implementation of a digitized packing scheme.

# DISCUSSION

## Implications for practical

A mixed integer planning model is constructed to solve the 3D-MODRPP and a 3D packing system with multiple bin sizes of open dimensions is proposed. In the model, a new method of assuming 0–1 integer variables is used to reduce the number of constraints, thus reducing the computational complexity of the model. A dataset for the 3D-MODRPP problem has also been generated based on the classical dataset for the single bin 3D packing problem, and superior results have been achieved. The practical impacts of this study are as follows:

1. Reduced complexity and computational time: Compared with the existing methods, the models and methods studied in this article can reduce the model complexity and computational time when solving 3D packing problems with multiple bin types. This will improve the efficiency of logistics companies in practice and reduce operational costs.

2. Improve resource utilization: The research presented in this article significantly contributes to optimizing bin dimensions, thereby maximizing bin space utilization while minimizing wastage. The findings of this study have valuable implications for reducing warehouse space requirements and enhancing overall resource utilization efficiency.

3. Drive transport efficiency: Optimized 3D packing solutions will help to reduce the volume of goods, thereby reducing wasted space in transport, increasing packing efficiency, and further improving transport efficiency.

4. Expand the scope of research: The research presented in this article offers a novel solution to the complex 3D packing problem involving multiple bin types within the context of e-commerce warehousing. By addressing this intricate challenge, the article significantly expands the boundaries of existing packing problem research. This expansion not only broadens the horizons of academic exploration but also holds the potential to revolutionize practical logistics optimization.

## Implications for theory

This study explores the solution of multiple bin 3D packing problems by applying it to the field of e-commerce warehousing. This provides a typical bin for the application of combinatorial optimization problems in different application areas, which is also of guiding significance for the application of problems in other areas. It is mainly reflected in the following aspects:

1. Algorithmic and optimization method innovation: In this article, we present a novel approach aimed at minimizing the count of 0–1 variables associated with overlapping constraints and rotational constraints. Our method achieves this by considering 0–1 integer variables, leading to a reduction in both constraint quantity and computational complexity within the model. This algorithmic advancement not only yields tangible outcomes in effectively addressing diverse challenges within 3D packing problems but also introduces fresh insights and methodologies applicable to a broader spectrum of combinatorial optimization issues.

2. Extension of application area: The research presented in this article opens up broader avenues and directions in the field of 3D packing. We aspire for these findings to serve as a catalyst, inspiring researchers to delve deeper into intricate packing challenges and thereby fostering the advancement of the field.

3. Dataset generation and verification of authenticity: The dataset produced within this article closely aligns with the challenges posed by the 3D-MODRPP problem, thereby furnishing a notably dependable resource for tackling real-world predicaments. Furthermore, the devised methodology for dataset generation and the meticulous process employed for validating outcomes hold considerable significance in the realms of dataset construction and ensuring credibility across diverse domains.

4. Combination of theoretical models and practical problems: This research synergistically integrates a mixed integer programming model with established hardware and software technologies, effectively addressing real-world challenges within the realm of e-commerce warehousing. This harmonious fusion of theoretical constructs and

pragmatic issues serves to enhance the symbiotic relationship between academia and industry, fostering collaborative efforts and driving the resolution of tangible problems in a concerted manner.

## CONCLUSIONS AND PROSPECT

In this study, we conducted an extensive investigation of the 3D bin packing problem, with a particular focus on the e-commerce warehousing domain involving multiple open bin sizes. Our research work culminated in the development of a novel 3D bin packing system capable of accommodating a wide range of open bin sizes. By formulating a hybrid integer planning model (3D-MODRPP) for the multi-bin 3D bin packing problem and introducing a new 0–1 integer variable assumption approach, we succeeded in reducing the complexity of the model and the computational time. The experimental results clearly show that the system significantly improves bin packing efficiency while minimizing space waste compared to existing methods. In addition, our study provides researchers with broader opportunities to explore this area.

To summarize the full text, the main contributions of this study include the following:

1. Building upon existing research, this article delves into the optimization of multiple-bin types within the realm of e-commerce. One notable research gap in the past has been the scarcity of investigations into the quandary of open-size 3D packing. Therefore, this study aims to bridge this gap by shedding light on this crucial aspect and expanding the boundaries of research in the field of 3D binning.

2. In this article, we present a novel approach to address the 3D-MODRPP using a hybrid integer programming model. Our proposed model takes overlap constraints and rotation constraints into account while simultaneously reducing the number of 0–1 variables. We introduce a new method that assumes 0–1 integer variables, resulting in a significant reduction in the number of constraints. This enhancement not only simplifies the complexity of the model but also leads to decreased computation time, ultimately enhancing the efficiency and usability of the system.

3. Based on the aforementioned hybrid integer planning model and leveraging well-established hardware and software technologies, this article introduces an advanced, fully automated digital intelligent bin packing system. This system guarantees the utmost precision in bin packing operations by conducting multiple reviews of order information, thereby enhancing accuracy. Moreover, it serves as a valuable resource and point of reference when dealing with intricate real-life scenarios.

However, this study has certain limitations that need to be addressed. First, the experimental data used in this study may be relatively simplified, and thus the model needs to be further validated and optimized for practical applications. Therefore, it is imperative that the model be further validated and optimized to ensure its applicability in real-world situations. Second, although the model proposed in this article shows favorable results in terms of reduced computation time and increased efficiency, there is still room for further improvement. For example, it is necessary to further refine the modeling of constraints and explore more effective solution algorithms to maximize its potential.

In the future, there are several ways to further enhance and extend the results of this study. First, it is recommended that the constraints of the model be strengthened to incorporate a wider range of limitations applicable to real-world situations. This may involve considering additional factors such as commodity characteristics, packing material costs, and other relevant variables. By integrating these considerations, the model will be able to produce more optimal solutions that are consistent with real-world constraints. Second, it would be beneficial to explore the integration of machine learning and artificial intelligence techniques to develop an intelligent 3D bin packing system. By harnessing the power of these advanced technologies, an intelligent 3D bin packing system can be built to improve the automation and decision accuracy of the system.

### Funding

This project is supported by the Beijing Wuzi University Youth Research Fund Project Name: Research on Intelligent E-commerce Unmanned Warehouse Packing and Loading Optimization Strategy and Green Recycling Mode; Project Number: 2023XJQN14, and by the Graduate Science and Technology Innovation Project of Capital University of Economics and Business (Project Name: Research on Goods Sorting on Conveyor Belts Based on Binocular Vision; Grant Number: 2023KJCX062). The funders had no role in study design, data collection and analysis, decision to publish, or preparation of the manuscript.

### Grant Disclosures

The following grant information was disclosed by the authors:
Beijing Wuzi University Youth Research Fund: 2023XJQN14.
The Graduate Science and Technology Innovation Project of Capital University of Economics and Business:  2023KJCX062.

### Competing Interests

The authors declare there are no competing interests.

### Author Contributions

- Jianglong Yang conceived and designed the experiments, performed the computation work, prepared figures and/or tables, and approved the final draft.
- Kaibo Liang conceived and designed the experiments, performed the computation work, prepared figures and/or tables, authored or reviewed drafts of the article, and approved the final draft.
- Huwei Liu conceived and designed the experiments, performed the computation work, authored or reviewed drafts of the article, and approved the final draft.
- Man Shan analyzed the data, prepared figures and/or tables, and approved the final draft.
- Li Zhou conceived and designed the experiments, prepared figures and/or tables, authored or reviewed drafts of the article, and approved the final draft.

- Lingjie Kong performed the experiments, authored or reviewed drafts of the article, and approved the final draft.
- Xiaolan Li analyzed the data, authored or reviewed drafts of the article, and approved the final draft.

## Data Availability

The data is available at GitHub and Zenodo:

– https://github.com/riverdragon2023/dataset.git.

– riverdragon2023. (2023). riverdragon2023/dataset: v1.0 (riverdragon). Zenodo. https://doi.org/10.5281/zenodo.8324824.

## Supplemental Information

Supplemental information for this article can be found online at http://dx.doi.org/10.7717/peerj-cs.1613#supplemental-information.

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
