# Peer review of "Optimizing e-commerce warehousing through open dimension management in a three-dimensional bin packing system"

_PeerJ Computer Science, doi:10.7717/peerj-cs.1613_

## Round 0.1 · original submission · Major Revisions

Dear authors,

Please revise the paper according to the comments of reviewers and provide replies point to point for each comment.

Reviewer 1 ·

Basic reporting

English writing is appropriate, clear and professional.
The background is well established. Reviewed literature is relevant and contemporary.
The paper is well-structured and well-written.
Writing in some of the figures is hard to read. Please try to enlarge the font and make it clear. Also, try to use the same font in all figures (it should also be the same as in the rest of the paper).
Raw data are supplied.

Experimental design

The subject of the paper is very interesting and in line with the aims and scope of the Journal.
The authors mentioned the research gap, but as late as in the conclusion. They should identify the main research gaps based on the literature that they reviewed.
The investigation is in line with the academic standards.
Methods are explained with enough detail, although the numeration of the equations is very confusing. Please try to simplify the enumeration.

Validity of the findings

The authors pointed out well the main novelties and contributions of their paper.
All underlying data are provided.
The paper does not have a proper discussion. The authors did not discuss how the results can be interpreted in the perspective of previous studies. Discussion should clearly and concisely explain the significance of the obtained results to demonstrate the actual contribution of the article to this field of research when compared with the existing and studied literature.
The authors mention implications in the introduction but then don’t provide any concrete managerial (practical) or theoretical implications.

Additional comments

Table captions are usually written above the table, not below.
The numeration of equations should be checked. For example, there are two equations numbered (20’’).
References in the reference list are not formatted consistently.
Some references are missing some information (such as volume, page numbers, etc.). Check all references.

·

Basic reporting

Good English is used throughout the paper. The literature followed the rest of the text well, although more references can be added to make the paper even better. The structure of the paper, tables and pictures is good, there are no complaints. There is a larger number of figures, but this is due to the very problem that this paper solves. The methodology and methods are well explained. The results are clear and well explained.

Experimental design

In the introduction, the authors clearly defined the goals of this research, but did not specify which gaps this research solves. It is necessary that they state this in the introduction. The research was very well done according to good technical standards. The ethical standards in the paper are well defined. Valid and the construction of the model is good.

Validity of the findings

When creating the model, the variables were clearly defined. This paper helps in the implementation of logistics and package delivery. Through the diameter it is very well explained. This research is useful for further similar problems. The results are well defined and it is shown how the orders should be distributed in order to optimize the packaging. The conclusions are well defined, it is only necessary to state what the limits of this research were.

Additional comments

In the previous comments, it was stated what the authors should correct in the paper in order to make it as good as possible.

---

## Round 0.2 · accepted · Accept

Dear authors,

Your paper has been accepted by both reviewers.

Reviewer 1 ·

Basic reporting

The authors have successfully addressed all issues from the previous review round.

Experimental design

The authors have successfully addressed all issues from the previous review round.

Validity of the findings

The authors have successfully addressed all issues from the previous review round.

Additional comments

The authors have successfully addressed all issues from the previous review round.

·

Basic reporting

The paper has been corrected in accordance with the comments

Experimental design

The paper has been corrected in accordance with the comments

Validity of the findings

The paper has been corrected in accordance with the comments

Additional comments

The paper has been corrected in accordance with the comments